# *p*-Aminophenylalanine Involved in the Biosynthesis of Antitumor Dnacin B1 for Quinone Moiety Formation

**DOI:** 10.3390/molecules25184186

**Published:** 2020-09-12

**Authors:** Xiaojing Hu, Xing Li, Yong Sheng, Hengyu Wang, Xiaobin Li, Yixin Ou, Zixin Deng, Linquan Bai, Qianjin Kang

**Affiliations:** 1State Key Laboratory of Microbial Metabolism, Joint International Research Laboratory of Metabolic & Developmental Sciences, and School of Life Sciences & Biotechnology, Shanghai Jiao Tong University, Shanghai 200240, China; dyhuxiaojing@163.com (X.H.); lixing9914@sjtu.edu.cn (X.L.); John.Sheng@sjtu.edu.cn (Y.S.); wanghengyu0203@163.com (H.W.); xiaobinli@sjtu.edu.cn (X.L.); yixinou@sjtu.edu.cn (Y.O.); zxdeng@sjtu.edu.cn (Z.D.); bailq@sjtu.edu.cn (L.B.); 2Zhuhai Precision Medical Center, Zhuhai People’s Hospital, Zhuhai Hospital Affiliated with Jinan University, Zhuhai 519000, China

**Keywords:** *Actinosynnema pretiosum*, comparative genomics, dnacin B1, antitumor, biosynthetic gene cluster, *p*-aminophenylalanine

## Abstract

*Actinosynnema* species produce diverse natural products with important biological activities, which represent an important resource of antibiotic discovery. Advances in genome sequencing and bioinformatics tools have accelerated the exploration of the biosynthetic gene clusters (BGCs) encoding natural products. Herein, the completed BGCs of dnacin B1 were first discovered in two *Actinosynnema pretiosum* subsp. *auranticum* strains DSM 44131^T^ (hereafter abbreviated as strain DSM 44131^T^) and X47 by comparative genome mining strategy. The BGC for dnacin B1 contains 41 ORFs and spans a 66.9 kb DNA region in strain DSM 44131^T^. Its involvement in dnacin B1 biosynthesis was identified through the deletion of a 9.7 kb region. Based on the functional gene analysis, we proposed the biosynthetic pathway for dnacin B1. Moreover, *p*-amino-phenylalanine (PAPA) unit was found to be the dnacin B1 precursor for the quinone moiety formation, and this was confirmed by heterologous expression of *dinV*, *dinE* and *dinF* in *Escherichia coli*. Furthermore, nine potential PAPA aminotransferases (APAT) from the genome of strain DSM 44131^T^ were explored and expressed. Biochemical evaluation of their amino group transformation ability was carried out with *p*-amino-phenylpyruvic acid (PAPP) or PAPA as the substrate for the final product formation. Two of those, APAT4 and APAT9, displayed intriguing aminotransferase ability for the formation of PAPA. The proposed dnacin B1 biosynthetic machinery and PAPA biosynthetic investigations not only enriched the knowledge of tetrahydroisoquinoline (THIQ) biosynthesis, but also provided PAPA building blocks to generate their structurally unique homologues.

## 1. Introduction

The genus *Actinosynnema* belongs to rare actinomycete and is closely related to *Nocardia* in phylogeny [1,2]. To date, two species including *Actinosynnema mirum* [1] and *Actinosynnema pretiosum* [3] with two subspecies were discovered in this genus. Although species in this genus are limited, *Actinosynnema*-origined natural products exhibit a number of biological and pharmacological activities, such as antitumor (ansamitocin [4,5,6,7], dnacin B1 [8] and actinosynneptides [9]), antibacterial (nocardicins [10]), anti-fungi (validoxylamine A [11]), metalloprotease inhibitor (propioxatin [12]), and siderophore (mirubactin [13]) (Figure 1A). Its promising application in pharmacy promoted the sequencing of the completed genome of four *Actinosynnema* strains (including two type strains) [14,15,16,17], which not only boosted the titer optimization of bioactive products by systematic metabolic engineering, but also facilitated the discovery of the *Actinosynnema*-derived antibiotics and dissection of their biosynthetic machineries.

Strain DSM 44131^T^ (= *A. pretiosum* subsp. *auranticum* ATCC 31309) is an aerobic actinobacterium and its mycelium displays the typical feature of the genus *Actinosynnema* [3]. The strain is of interest due to its production of potent anti-tumor agents ansamitocin and dnacin B1. The chemical structure of dnacin B1 contains an intricate polycyclic architecture, which features the characteristic of naphthyridinomycin (NDM) class among the tetrahydroisoquinoline (THIQ) alkaloids [18]. The THIQ alkaloids display strong antitumor and a broad spectrum of antimicrobial activities [19], which are attributed to their ability to alkylate the DNA in the minor groove. The remarkable efficacy of anticancer application has been proved by the marketing of ecteinascidin 743 (ET-743) [20]. The THIQ alkaloids fascinate both biological and chemical studies by virtue of their complex chemical structures, complicated biosynthetic and biochemical pathways, and their pharmacological potential.

The biosynthesis of THIQ antibiotics has been intensively investigated by classical precursor feeding experiments, as well as genetic and biochemical approaches. The biosynthetic pathway of natural THIQ consists of the unified iterative non-ribosomal peptide synthase (NRPS) for Pictet-Spengler (PS) reaction, non-proteinogenic amino acid building blocks, a cryptic fatty acyl chain or a leader peptide for NRPS, a hydroxyethyl moiety from ketose, and a powerful terminal reductase domain embedded in the C-terminal of last NRPS module [21]. In addition, the biosynthetic investigations on THIQ alkaloids had demonstrated that the quinone units of ET-743 [22], saframycin A (SFM-A) [23], safracin B (SAC-B) [24], NDM [25] and quinocarcin (QNC) [26] were originated from tyrosine or phenylalanine. To date, dnacin B1 represented a unique structure within THIQ alkaloids for its amino-substituted quinone moiety. However, the BGC of dnacin B1 and its biosynthetic mechanism are still elusive.

In this research, we present the results of the comparative genome mining within four high-homology *Actinosynnema* strains, leading to the discovery of dnacin B1 BGC from strain DSM 44131^T^ and strain X47. Targeted disruption of a 9.7 kb NRPS encoding DNA region in the chromosome of strain DSM 44131^T^ confirmed that BGC was responsible for dnacin B1 biosynthesis. In addition, we focus on the analysis of the genetic basis for PAPA, which was the precursor of dnacin B1 for the quinone moiety, and the heterologous expression of PAPA was achieved in *E. coli*. Moreover, the aminotransferases responsible for the amino group transformation in the mature PAPA were systematically explored and evaluated. This research presented the understanding of the biosynthesis mechanism of dnacin B1 and enriched the building blocks of THIQ antibiotics for creating new bioactive compounds.

## 2. Results

### 2.1. Genome Sequencing Revealed a Circular Chromosome of Strain DSM 44131^T^

The rapid development in genome sequence technology has enhanced the blooming of genome sequencing for various microorganisms and promoted the development of a number of bioinformatics tools for natural product exploration [27,28]. In order to deeply explore the secondary metabolite potential of *A. pretiosum* subsp. *auranticum*, the high-quality draft genome of strain DSM 44131^T^ was sequenced through Illumina Miseq technology, and the acquired reads were assembled by A5-miseq. The genome sequence revealed a circular chromosome of 8,105,537 bp, a G+C content of 73.95%, and 6,663 protein coding genes. Total 7 rRNA and 57 tRNA genes were predicted as well (Appendix A). The information of each predicted gene, such as gene product annotation and KEGG orthology, were available on the GeneBank database. The function analysis of the predicted proteins was based on COGs categories using the COGs database. The categories assigned a part of genes to transcription (9.37%), carbohydrate transport and metabolism (5.63%), amino acid transport and metabolism (4.91%), signal transduction mechanisms (4.55%), inorganic ion transport and metabolism (4.05%), energy production and conversion (3.94%), and secondary metabolites biosynthesis, transport and catabolism (3.03%). While a large number of genes were assigned to unknown function category (31.51%) (Appendix A). The genome sequence data have been deposited in the GenBank database and the accession number is JABBHD000000000.1; the BioProject number is PRJNA541432.

### 2.2. Comparative Genomic Analysis within Actinosynnema Resulted in the Discovery of Dnacin B1 BGC

Comparative genomics has not only been used in the evaluation of the evolutionary relationship between different strains, but also in the prediction of the putative functional genes in many strains [29,30]. The online antiSMASH analysis of the harbored BGCs within chromosomes of these four strains indicated that strain DSM 44131^T^ was nearly identical to strain X47 (CP023445.1), and different from strain DSM 43827^T^ (CP001630.1) and strain ATCC 31280 (CP029607.1) (Appendix A), in which strain DSM 44131^T^ contained 28 potential BGCs, which was the average number of typical streptomycete strains (Appendix A). In addition, the 66.9 kb DNA fragment garnered our significant interest, given that it was located in both strain DSM 44131^T^ and strain X47, close to a NRPS BGC, and exhibiting 64% similarity with the biosynthetic gene cluster of the THIQ alkaloid, naphthyridinomycin (NDM) (Figure 1B). This discovery showed us that this region should be related to the biosynthesis of two THIQ alkaloids, dnacin A1 and dnacin B1, which were isolated from strain DSM 44131^T^ (identified as *Nocardia* sp. No. C-14482 before) [8].

Subsequently, the crude extract of strain DSM 44131^T^ was prepared, as well as the initial HPLC-MS analysis, however, it failed to detect the production of dnacin. To facilitate the following study, we tried to improve dnacin yield through media optimization and supplement with resin HP-20, and the titer was evaluated by the transcription levels of relevant genes with qPCR analysis and the inhibition activity against *S. aureus* ATCC 25923. A set of optimizations of the fermentation conditions in four different media (including H, K, 17 and J medium) were carried out. Fortunately, J medium showed a preferred expression level of the targeted functional genes, which might be due to a better growth level or some other unknown reasons caused by resin HP-20 absorption (Appendix A). The HPLC-MS analysis displayed that dnacin B1 was clearly produced in strain DSM 44131^T^, fermented in J medium with resin HP-20 for three days (Figure 2D,E).

### 2.3. Functional Genes Involved in the Assembly of Dnacin B1 Skeleton

The detailed bioinformatics analysis of the dnacin B1 BGC from strain DSM 44131^T^ revealed that it contained 41 open reading frames encoding for NRPS, PAPA NRPS extender unit, Pictet–Spengler (PS) reaction, tailoring modification, resistance and regulations (Figure 1C) [21,31,32]. The putative function of the related genes was listed in Table 1. To verify the function of four modular NRPS genes (*dinL*, *dinM*, *dinN* and *dinO*) in dnacin B1 biosynthesis, a dnacin B1 BGC inactivation (Δ*din*) mutant HXJA01 was constructed by deleting these four genes and replaced with an apramycin resistance gene. HPLC-MS analysis displayed that no dnacin B1 was produced in mutant HXJA01 compared with the wild type strain, indicating the BGC was responsible for the biosynthesis of dnacin B1 in strain DSM 44131^T^ (Figure 2).

In dnacin B1 BGC, the modular NRPS genes *dinL*, *dinJ, dinN* and *dinO* displayed about 52% sequence similarity with those located in NDM BGC. The enzymes encoded by these four genes were responsible for a typical tetrahydroisoquinoline NRPS biosynthesis system (Figure 3) [21]. DinN was a didomain enzyme with acyl-CoA ligase (AL) and peptidyl carrier protein (PCP) activities, responsible for the cryptic fatty acyl chain loading and initiation of the dnacin B1 skeleton assembly. DinO was similar to the typical NRPS proteins composed of three C-A-PCP modules, and all of the A domains recognized alanine as the extension substrate.

The two-component transketolases encoded by *dinS* and *dinU* were homologous to *napB* and *napD* in the NDM BGC, supporting their functions related to the generation of hydroxyethyl unit from d-xylutose-5-phosphate and transferred to DinT [33]. DinQ resembled a family of KASIII proteins that served as acyl unit-loading activity, which was probably involved in tethering glyceryl group to the 4′-phosphopantetheinyl arm of the ACP of DinT. The A domain of DinL (46% identity with A domain in NapL) was inactive, whereas the C domain was in charge of the condensation of the hydroxyethyl group and the nascent *N*-fatty acid substituted peptidyl chain, providing the tethered PCP with a linear thioester intermediate.

DinJ played an important role as NRPS in THIQ biosynthesis. It comprised a tetradomain including a reductase (RE) domain embedded at the terminal of the C-A-PCP, which catalyzed PS- and Mannich-type reactions, conferring the core scaffolds in dnacin B1 biosynthesis [26]. The 3-OH-PAPA precursor was transferred to the PCP domain in DinJ after the adenylation by DinH. The A domain of DinJ (69% indentity with A domain of NapJ) recognized and activated *L*-(*E*)-4,5-dehydroarginine, which was modified by DinI (non-heme iron, α-ketoglutarate-dependent oxygenase). Subsequently, DinJ-RE domain reduced the assembling hybrid-peptidyl chain anchored at the PCP domain at DinL to release the aldehyde intermediate. Then, the DinJ-C domain catalyzed the aldehyde intermediate and the PCP linked aromatic acid through PS reaction. The DinJ-RE domain further mediated the reduction in the DinJ-PCP-linked product to deliver an aldehyde intermediate, preparing for the intramolecular Mannich reaction with another DinJ-PCP-tethered *L*-(*E*)-4,5-dehydroarginine substrate. The resultant DinJ-PCP attached intermediately, then underwent reduction and cyclization to form the final aldehyde product.

In addition to the NRPS skeleton assembly and tetrahydroisoquinoline scaffold formation, a set of post-tailoring enzymes encoded by *dinX*, *dinY*, *dinC* and *dinA* were present in dnacin B1 BGC. Two tailoring enzymes DinX and DinY were predicted to proceed with the hydroxylation and *N*-methylation of dnacin B1 skeleton. The gene product of *dinX* showed 67% sequence similarity with the FAD-dependent monooxygenase NapA, which might be responsible for the hydroxylation in naphthyridinomycin biosynthetic pathway. DinY showed 65% sequence similarity with the methyltransferase NapV, which might be in charge of the *N*-methylation in dnacin B1 biosynthetic pathway. The peptidase DinC showed 56% similarity with NapG, which might be the membrane-bound peptidase participating in the cleavage of leader peptide and then exporting the precursor from the cell. DinA, an FAD-linked oxidase with 75% similarity with NapU, might have the same function as NapU, which was confirmed to be in charge of oxidative activation and over the oxidative inactivation of a matured prodrug [31]. The dnacin B1 BGC contained three regulator genes, separately belonging to the TetR family (*dinR2*), SARP family (*dinR3*) and LysR family (*dinR9*). DinR8, annotated as a DNA excision repair enzyme UvrA, showed 84% similarity with NapR1, and was deduced as the resistance gene in dnacin B1 biosynthesis [34]. In addition, five ABC transporter genes, *dinR1* and *dinR4-7* were also found in dnacin B1 BGC (Table 1).

### 2.4. Reconstitution of PAPA Pathway Revealed TyrB Participating in the Mature of PAPA

In the dinacin B1 biosynthetic machinery, DinV resembled the proteins from aminodeoxychorismate synthase (ADCS) family that are widely found in *Actinobacteria*, and was involved in the dehydration of C-5 hydroxyl group and transferring an amino group into chorismic acid at C-3 to generate of *p*-amino-deoxychorismate [35]. This was further proved by a phylogeny comprising ADCS, anthranilate synthase (AS), isochorismate synthase (IS) and salicylate synthase (SS) (Appendix A). Genes *dinE* and *dinF* shared a homology with 4-amino-4-deoxychorismate mutase *papB* and dehydrogenase *papC* from *Streptomyces venezuelae*, respectively, and might be candidates for the transformation of enol pyruvic acid and decarboxylation to generate *p*-amino-phenylpyruvic acid [36]. Finally, an aminotransferase of the TyrB family, whose encoding gene was located apart from the BGC of dnacin B1, mediated an amino group transformation at the keto position to produce PAPA. The PAPA biosynthetic machinery in strain DSM 44131^T^ was also considered to be a pathway parallel to that from chorismic acid to tyrosine (Appendix A).

In order to reconstitute the originated PAPA pathway in dnacin B1 BGC, *dinV* was cloned from gDNA of strain DSM 44131^T^ and inserted into the vector pETDuet to obtain pJQK354, *dinE* and *dinF* were amplified and inserted into vector pCDFDuet to afford pJQK355. Each gene was under the control of its own T7 promoter and transcription ended by the terminator. These two recombinant plasmids were together transferred into *E. coli* BL21 (DE3), conferring HXJE03. Meanwhile, both pETDuet and pCDFDuet were transferred into *E. coli* BL21 (DE3) generating HXJE04 as the control strain. Then, the fermentation was carried out in modified M9 medium with corresponding antibiotics and induced by IPTG at 25 °C and 220 rpm. After the fermentation was completed, HPLC-MS analysis of both fermentation broth showed that PAPA was clearly produced in HXJE03, while there was no PAPA detected in strain HXJE04 (Figure 4). In the absence of aminotransferase that catalyzed the last mature step of PAPA, the achievement of the PAPA production in strain HXJE03 suggested that the endogenous aminotransferases might have participated in the last amino transformation, especially for the aminotransferase involved in the aromatic amino acid biosynthesis. This deduction was reasonable because PAPA has a similar chemical structure to tyrosine, and as we know, the aminotransferase TyrB catalyzed the last step of amino group transformation in the mature tyrosine. In order to evaluate the enzymatic property of aminotransferase in PAPA biosynthesis, gene *tryB*, responsible for the amino transformation in tyrosine biosynthesis, was cloned from the gDNA of *E. coli* MG1655 and heterologous expressed in *E. coli* BL21 (DE3). As was expected, incubation of the recombinant N-His_6_-TyrB with PAPP and glutamate (Glu) as the substrates generated PAPA, whereas PAPP also could be produced in the presence of PAPA and α-ketoglutarate (α-KG) as substrates (Figure 5). The chemical structures of PAPA from the TyrB-catalyzed reactions were determined by NMR and HR-ESI-MS analysis (Appendix A). This result indicated that the TyrB involved in the tyrosine biosynthesis displayed a flexible substrate and transferred an amino group in a reversible reaction with PAPP or PAPA and related auxiliary substrates.

### 2.5. Aminotransferases Responsible for the PAPA Maturation Dispersed within the Genome

Aiming at exploring the possible aminotransferase participating in PAPA biosynthesis during dnacin B1 assembly, nine candidate aminotransferase (AT) genes *apat1*-*9* were screened from the annotation results of the whole genome of strain DSM 44131^T^ with the reference sequence of TyrB (Appendix A). To date, the aminotransferases have been divided into five classes, among which tyrosine aminotransferase (TyrB) belongs to class I, together with aspartate aminotransferase, alanine aminotransferase and other aromatic aminotransferase [37]. Moreover, class I is further divided into Ia, Ib and Ic subgroups [38,39]. To better confirm the function of the nine aminotransferases APAT1-9, the phylogenetic relationship was analyzed among these candidates and the other related class I aminotransferases (Appendix A). Consequently, TyrB was located in the Ia subgroup, while all of the nine candidates, together with some extremophile-originated aminotransferases, gathered into the Ib subgroup. This result implied that the unusual function or property might be involved in the strain DSM 44131^T^-originated aminotransferases.

To further correlate the function of the nine aminotransferases APAT1-9 in vitro in the biosynthesis of PAPA from PAPP or the reversed reaction, the encoded genes were amplified from the gDNA of strain DSM 44131^T^ and cloned into pET28a, respectively. The obtained plasmids were individually transferred into *E. coli* BL21 (DE3), and all of the N-terminally His_6_-tagged fusion proteins were obtained after purification by Ni^2+^ affinity chromatography. For systematic comparison of their catalytic abilities, equal amounts of APAT1-9 and TyrB were used to convert PAPP to PAPA using Glu as amino donors, or convert PAPA to PAPP with α-KG as an amino acceptor, individually at optimal reaction conditions. The reactions were subjected to HPLC for the yield analysis of the final products with authentic standards. Remarkably, APAT4 and APAT9 showed a closer relationship in the phylogenetic tree and displayed the preferable conversion rate of 63% and 74% for PAPA formation using PAPP as the amino group acceptor, which were superior to TyrB catalyzed ability. APAT3, APAT6, APAT7 and APAT8 showed the moderated conversion rates (less than 30%) during PAPA maturation with PAPP as the substrate. However, APAT1, APAT2 and APAT5 only exhibited catalytic capacity for the PAPP formation with PAPA as the substrate (Figure 5). Our systematic investigation of the aminotransferases involved in PAPA biosynthesis from the genome of strain DSM 44131^T^ present the preferred aminotransferase candidates for further heterologous expression of PAPA in other hosts.

## 3. Discussion

Nowadays, great efforts have illuminated the mechanisms underpinning the biosynthesis of a diverse array of natural products from various bacteria. This research makes it possible to link the known natural products to their biosynthetic gene clusters. Moreover, the development of genome sequencing technology and many computational tools for natural product discovery have promised a remarkable increase in mining bacterial cryptic BGCs [40]. The available information on the genomic sequences of *Actinosynnema* revealed their circular chromosome structures with a relatively high GC content (>73%) among *Actinobacteria*. The initial comparative genome analysis of four sequenced *Actinosynnema* strains exhibited extremely high homology with the almost identical multilocus sequences by phylogenetic marker analysis. However, detailed analysis by Antismash revealed that chromosome harbored diversified secondary metabolite BGCs. Based on the comparative information of the BGCs of *Actinosynnema* species, we found the gene cluster responsible for dnacin B1 biosynthesis in strain DSM 44131^T^, which was not reported before.

The diversity of THIQ structures resulted mainly from different kinds of amino acid precursors for the recognition of NRPS proteins and diverse post-modifications. Previous incorporation experiments with labeled precursor and biosynthetic investigations demonstrated that the non-proteinogenic aromatic amino acid was responsible for the quinone moiety of THIQ and originated from tyrosine in ET-743, SFMs, SAC-B, NDM and phenylalanine in QNC. The bioinformatics analysis showed that DinV, DinE and DinF individually encoded 4-amino-4-deoxychorismate synthase, 4-amino-4-deoxychorismate mutase and 4-amino-4-deoxyprephenate dehydrogenase in dnacin B1 BGC, and displayed 48%, 34% and 40% similarities with PapA, PapB and PapC in *S. venezuelae*, which were responsible for the biosynthesis of PAPA in chloramphenicol. Thus, it was deduced that, similarly to the chloramphenicol biosynthesis, the quinone moiety of dnacin B1 arose from PAPA rather than tyrosine. In addition, the biosynthetic pathway reconstitution of PAPA in *E. coli* suggested that TyrB was responsible for the transformation of amino group into PAPP, although the detailed biochemical analysis of TyrB in PAPA assembly was unclear. Our systematic screening of the aminotransferases involved in PAPA biosynthesis from the genome of strain DSM 44131^T^ resulted in the discovery of APAT4 and APAT9, displaying intriguing amino transformation ability, which might be a preferred aminotransferase candidate for further heterologous expression of PAPA in other hosts. All the screened APAT genes, including *apat4* and *apat9,* were found in the other three *Actinosynnema* strains regardless of the production of dnacin B1 or PAPA (Appendix A). This appearance revealed that the biosynthesis of PAPA does share the same aminotransferases as other aromatic amino acids in basic metabolism.

The biosynthetic pathway of PAPA can be verified by the co-occurrence of DinV, DinE and DinF in a heterologous host *E. coli* BL21 (DE3), which makes it possible to explore more PAPA pathways containing BGCs by MultiGeneBlast. Our genome mining results displayed that homologs of PAPA biosynthetic gene cassettes were discovered in several *Streptomyces* species (Appendix A), as well as in *Saccharopolyspora* species (Appendix A). Not all the aminotransferase genes involved in PAPA biosynthesis were anchored in these BGCs. Remarkably, detailed analysis of these explored BGCs revealed that the encoding metabolites were distributed into non-ribosomal peptides and other hybrid types. This finding reveals that PAPA biosynthesis genes would be the useful baits for discovering novel antibiotics using bioinformatics analysis.

## 4. Materials and Methods

### 4.1. Bacterial Strains, Plasmids and Reagents

All the bacterial strains and plasmids used in this research were summarized in Appendix A. All the chemical, biochemical, enzymes and other molecular biological reagents were purchased from standard commercial sources. All the primers used in this research are listed in Appendix A.

### 4.2. General Experimental Procedures

NMR spectra of PAPA were acquired in D_2_O using a Bruker AVANCE III-600 spectrometer (Bruker Corp., Germany) with tetramethylsilane (TMS) as an internal standard. HR-ESI-MS data were recorded using an Agilent 1290 Infinity II/6545 QTOF LC/MS instrument (Agilent Technologies, Santa Clara, CA, USA), with an Agilent TC C18 column (4.6 × 150 mm, 5 μm). The purification of PAPA was carried on Agilent 1290 Infinity II/1292 Prep FC instrument with an Agilent Eclipse XDB-C18 column (9.4 × 250 mm, 5 μm). Column chromatography separation for dnacin B1 detection was carried on MCI gel CHP20/P120 (Mitsubishi, Tokyo, Japan).

### 4.3. DNA Sequencing for Strain DSM 44131^T^

The DNA sequencing of strain DSM 44131^T^ was performed at Shanghai Personal Biotechnology Co., Ltd. The gene sequence assembly was using A5-miseq v20150522. Genes in the assembled sequence were predicted by GeneMarks (http://exon.gatech.edu/GeneMark/, version 4.32 April 2015) [41]. tRNA, rRNA and ncRNA were identified using tRNAscan-SE (version 1.3.1), Barrnap (0.9-dev) (http://github.com/tseemann/barrnap) and Rfam database, respectively. Protein-coding sequences were annotated based on BLASTP searches against NCBI NR, COG and KEGG databases with an E-value cut-off of 1e-6.

### 4.4. Bioinformatics Analysis

Three *Actinosynnema* genome sequences (*A. pretiosum* subsp. *pretiosum* ATCC 31280, *A. mirum* DSM 43827^T^, *A. pretiosum* subsp. *aurantium* X47) were downloaded from the NCBI GenBank database for the genomic comparative analysis along with strain DSM 44131^T^. The pair-wise alignment was done by MAUVE [42]. The secondary metabolites’ gene clusters contained in all the genomes were predicted using antiSMASH (https://antismash.secondarymetabolites.org/#!/start) [43]. The *orfs* in dnacin B1 biosynthetic gene cluster (dnacin B1 BGC) were analyzed using FramePlot 4.0 beta program (http://biosyn.nih.go.jp/2ndfind/) [44]. The corresponding proteins were compared with other known proteins using BLAST (https://blast.ncbi.nlm.nih.gov/Blast.cgi). The prediction of amino acid specificities of A domains was performed on the PKS/NRPS analysis website (http://nrps.igs.umaryland.edu/) [45].

To mine the potential PAPA specific aminotransferase, the whole genome sequence of strain DSM 44131^T^ was analyzed by KEGG (https://www.genome.jp/kegg/). Seven tyrosine aminotransferase homologies were found in the KEGG annotation. Another two genes were found through local blast analysis of strain DSM 44131^T^ genome with the sequence of an aminotransferase gene *tyrB* originated from *E. coli* as a reference.

### 4.5. Construction of the Dnacin B1 BGC Inactivation Mutant

Homologous recombination was used to construct the Δ*din* mutant strain in this research. As for the Δ*din* mutant strain, two pairs of primers din-del-L-F/R and din-del-R-F/R were used to amplify the left and right 1.7 kb/1.5 kb homologous arms with the gDNA of strain DSM 44131^T^ as template. The sequenced homologous arms were digested by *Eco*RI/*Hin*dIII and *Hin*dIII/*Kpn*I, following purified by gel purification kit, respectively. The two arms were inserted into *Eco*RI/*Kpn*I double-digested pJTU1278 to produce pJQK301. Then, an apramycin resistance gene cassette was inserted into the *Hin*dIII digested pJQK301 at the middle of the two arms to yield the pJQK302. The plasmid pJQK302 was transferred into *E. coli* ET12567/pUZ8002, and cultured in LB broth containing 50 μg/mL kanamycin, 25 μg/mL chloramphenicol and 50 μg/mL apramycin for conjugation. The *E. coli*-*Actinosynnema* bi-parental conjugation was performed for transformation of the pJQK302 into strain DSM 44131^T^ [46]. The double-crossover exconjugants named HXJA01 were screened by resistance and confirmed by PCR with primers din-val-F/R and din-in-F/R.

### 4.6. Optimization of Fermentation Medium

Four different fermentation mediums (H, K, 17 and J) were chosen to compare the production of dnacin B1. All ingredients were shown as the following. H medium: yeast extract 10 g/L, corn starch 20 g/L, glucose 5 g/L, glycerol 40 g/L, K_2_HPO_3_ 0.5 g/L, FeSO_4_·7H_2_O 0.2 g/L, CaCO_3_ 5 g/L, pH 7.4. K medium: yeast extract 8 g/L, malt extract 10 g/L, sucrose 15 g/L, soluble starch 25 g/L, MgCl_2_ 0.2 g/L, pH 7.4. 17 medium: glucose 10 g/L, soybean flour 30 g/L, corn steep liquor 10 g/L, glycerol 5 g/L, yeast extract 5 g/L, sucrose 10 g/L, NaCl 5 g/L, CaCO_3_ 2 g/L, MgCl_2_ 0.2 g/L, pH 7.4. J medium: dextrin 50 g/L, corn steep liquor 30 g/L, (NH_4_)_2_SO_4_ 1 g/L, CaCl_2_ 10 g/L, CaCO_3_ 5 g/L, pH 7.4.

The wild type strain DSM 44131^T^ was grown on a YMG (yeast extract 4 g/L, malt extract 10 g/L, glucose 4 g/L, pH 7.2) agar plate at 28 °C for activation. The mycelium was inoculated into 50 mL TSBY (tryptone soybean broth 30 g/L, yeast extract 10 g/L, sucrose 103 g/L, pH 7.2) in 250 mL shake-flasks as seed cultures at 28 °C and 220 rpm for 24 h on a rotary shaker. For fermentation, 5 mL of the seed culture was inoculated into 100 mL fermentation broth in 500 mL shake-flasks at 28 °C and 220 rpm. After being cultured for 48 h, each medium was taken in half bottles to add HP20 resin of 6% (*w*/*v*) of fermentation volume.

### 4.7. Total RNA Extraction and q-PCR Analysis

Cells were harvested at 96 h of fermentation to extract total RNA according to the manufacturer’s protocol using Redzol reagent and SiMax membrane spincolumns (SBS Genetech; Shanghai, China). The gDNA was removed using DNaseI (Fermentas, Vilnius, Lithuania) for 4 h at 37 °C before reverse transcription. The first strand of cDNA was synthesized using the RevertAid H Minus First Strand cDNA Synthesis Kit (ThermoFisher, Waltham, MA, USA). All the experiments were performed in triplicate.

The expression levels of the dnacin B1 BGC in various fermentation broth were estimated by quantitative real-time PCR. Five genes (*dinI*, *dinP*, *dinQ*, *dinX* and *dinY*) located in different operons of the dnacin B1 BGC were chosen, along with the housekeeper gene *hrdB* as the control. The gene-specific primers were list in Appendix A. The quantitative real-time PCR reaction was performed on Thermo QuantStudio 3 Real-Time PCR System with Maxima™ SYBR Green/ROX qPCR Master Mix (ThermoFisher, USA) as follows: initial denaturation at 94 °C for 5 min followed by 32 cycles of denaturation at 94 °C for 30 s, annealing at 58 °C for 30 s, and extension at 72 °C for 30 s, with a final extension at 72 °C for 10 min. Finally, 2 μL products were subjected to run electrophoresis on 1.5% (*w*/*v*) agarose gel.

### 4.8. Production and Detection of Dnacin B1 in Strain DSM 44131^T^

For dnacin B1 detection, 2 L fermentation broth was harvested and centrifuged at 5000 rpm for 15 min. After being filtered, the supernatant was combined and fractionated by MCI gel eluted with gradients of 10% to 100% methanol. Each fraction was evaporated under reduced pressure at 40 °C. The fraction was dissolved in 1 mL methanol, and tested by the bioassay experiments using *Staphylococcus aureus* ATCC 25923 as the indicator strain for screening of the dnacin B1-containing candidates. The samples with a larger inhibition zone against *S. aureus* ATCC 25923 were combined and filtered for HPLC-MS analysis with an Agilent TC C18 column. The column was equilibrated with 95% solvent A (H_2_O, 0.1% formic acid) and 5% solvent B (methanol), and an analytic method was developed with the following program at a flow rate of 0.4 mL/min and UV detection at 280 nm: 0–25 min, a linear gradient from 5% B to 60% B; 25–45 min, a linear gradient from 60% B to 100% B; 45–53 min, constant 100% B; 53–54 min, a linear gradient from 100% B to 5% B; 54–60 min, constant 5% B. The compound with a retention time at 28 min showed an [M + H]^+^ ion at *m/z* = 389.1860 on a mass spectrometer, which was consisted with the molecular weight of dnacin B1 (C_19_H_24_N_4_O_5_, Exact Mass: 388.1747).

### 4.9. Reconstruction of PAPA Biosynthesis Pathway in E. coli

Three genes *dinV*, *dinE* and *dinF* were amplified by PCR using gDNA of strain DSM 44131^T^ as a template, with primers dinV-F/R, dinE-F/R and dinF-F/R, respectively. Gene *dinV* was inserted in *Bam*HI and *Hin*dIII sites of the pETDuet vector to construct pJQK354, while *dinE* and *dinF* was inserted, respectively, in *Bam*HI/*Hin*dIII and *Nde*I/*Xho*I sites of pCDFDuet vector successively to construct pJQK355. The two recombinant plasmids were co-introduced into *E. coli* BL21 (DE3) and screened by resistance and confirmed by PCR. Finally, the *dinV*, *dinE* and *dinF* heterologous expression strain HXJE03 was obtained.

Then, strain HXJE03 was cultured in LB broth containing 50 μg/mL streptomycin, 100 μg/mL ampicillin at 37 °C and 220 rpm overnight, and inoculated at a ratio of 2% (*v*/*v*) in 500 mL flasks containing 100 mL fresh modified M9 medium at 37 °C and 220 rpm [47]. When the optical density reached about 0.6 at 600 nm, 0.5 mM IPTG was supplemented into the broth and the cells were grown at 25 °C and 220 rpm for another 48 h. An equal volume of methanol was added into the fermentation broth to break the cells. After being centrifuged at 12,000 rpm at 4 °C, the supernatant was filtered and analyzed by HPLC-MS on an Agilent TC C18 column. The column was equilibrated with 98% solvent A (H_2_O, 0.1% formic acid) and 2% solvent B (methanol), and an analytic method was developed with the following program at a flow rate of 0.4 mL/min and UV detection at 245 nm: 0–10 min, constant 2% B; 10–12 min, a linear gradient from 2% B to 50% B; 12–17 min, constant 50% B. The compound with a retention time at 7.2 min showed a [M + H]^+^ ion at *m/z* = 181.0970 on a mass spectrometer, which consisted of the molecular weight of PAPA (C_9_H_12_N_2_O_2_, Exact Mass: 180.0899).

### 4.10. Protein Expression and Purification 

All nine PAPA aminotransferase genes from strain DSM 44131^T^ along with *tyrB* from *E. coli* were cloned with corresponding primers (Appendix A), and inserted in the *Nde*I and *Eco*RI sites of the pET28a vector. The recombinant plasmids were then transferred into *E. coli* BL21 (DE3). The final strains were individually inoculated into 5 mL LB medium containing 50 μg/mL kanamycin and incubated at 220 rpm, at 37 °C overnight. Then, 1 mL seed cultures were transformed into 2 L flasks containing 500 mL fresh LB medium incubated at 37 °C, until OD_600_ was reached at about 0.6, then 0.5 mM IPTG was supplied, and incubated another 20 h at 16 °C.

To purify the recombinant proteins, cells were harvested by centrifugation at 5000 rpm, 4 °C for 25 min, then the cells were resuspended in a 50 mL binding buffer (500 mM NaCl, 20 mM Tris, 5 mM imidazole, pH 7.9). After being sonicated on ice, the supernatants were collected by centrifugation at 10,000 rpm and at 4 °C for 40 min. Before loading onto Ni-NTA column (GE Healthcare, Ni Sepharose 6 Fast Flow), the supernatant was filtered through 0.45 μm filters. After being equilibrated with 20 mL binding buffer, the samples were eluted with washing buffer (binding buffer containing 100 mM imidazole) to obtain relatively pure *N*-His_6_-tagged recombinant proteins. Then, the purified proteins were concentrated and desalted by centrifugation at 3000 rpm at 4 °C using Amicon^®^ Ultra-15 centrifugal filter (Millipore) with TGE buffer (50 mM Tris, 0.5 mM EDTA, 50 mM NaCl, 5% glycerol, pH 7.9). Finally, all proteins were analyzed by SDS-PAGE and quantified by nano-drop (Figure 5A).

### 4.11. Enzyme Activity Analysis

The enzymatic properties of recombinant-purified TyrB and APATs were performed according to previous research [48]. In this study, both PAPA and PAPP were used as the substrates for the reactions and the reverse to test the reversible activity of corresponding enzymes.

For the overall comparison of the aminotransferase activities of nine APATs and TyrB in vitro, equal amounts of enzymes were used in the 50 μL reaction mixture, containing 50 mM Tris-HCl (pH 7.0), 100 μM PLP, 0.5 mM PAPP as an amino acceptor and 5 mM Glu as an amino donor. For the reverse reaction, 0.5 mM PAPA was used as an amino donor and 5 mM α-KG as an amino acceptor. The reaction was initiated by adding 70 μM enzyme, and incubated for 40 min at 37 °C. Then, an equal volume acetonitrile was added to abort the reaction. After being centrifuged at 12,000 rpm, 4 °C for 15 min, the supernatants were detected by HPLC. All reactions were performed in triplicate. For the purification of PAPA via enzyme reaction by TyrB, an amplified reaction system of 50 mL volume was processed. The final enzyme reaction mixture was separated by preparative HPLC with a 2:98 ration of methanol and 0.1% formic acid H_2_O at 1.8 mL/min to yield PAPA (*t*_R_ = 8 min, 2.2 mg).

## 5. Conclusions

In summary, on the basis of genome sequence of strain DSM 44131^T^, comparative genomic analysis plus with online AntiSmash analysis was carried out among the species in *Actinobacteria*. Detailed analysis of the diversified BGCs located in the genome of *Actinobacteria* resulted in the discovery of dnacin B1 BGCs in the genomes of strain DSM 44131^T^ and strain X47. Bioinformatics analysis indicated that forty-one genes were composed in the 66.9 kb cluster, including three genes for PAPA synthesis, four NRPS genes, and genes putatively involved in extension unit biosynthesis, modification, regulation, transport and resistance. The proposed biosynthetic pathway of dnacin B1 revealed PAPA as the aromatic precursor for quinone moiety formation. The biosynthetic cassette of PAPA containing of *dinV, dinE* and *dinF* was constructed, and there was successful heterologous expression in *E. coli*. In order to explore the PAPA aminotransferase involved in PAPA biosynthesis in the strain DSM 44131^T^, nine candidates of APAT1-9 were cloned, and soluble proteins were obtained. The biochemical identification revealed that APAT4 and APAT9 exhibited fascinating amino transformation ability compared to TyrB, which originated from *E. coli*. This research not only enriched the repertoire of THIQ biosynthetic investigations but also provides the superior gene elements for the construction of the PAPA biosynthetic pathway. More attractively, target-directed genome mining using a PAPA biosynthetic cassette as a probe promises a renaissance of the accelerated discovery of novel PAPA-derived natural antibiotics.

## Figures and Tables

**Figure 1 molecules-25-04186-f001:**
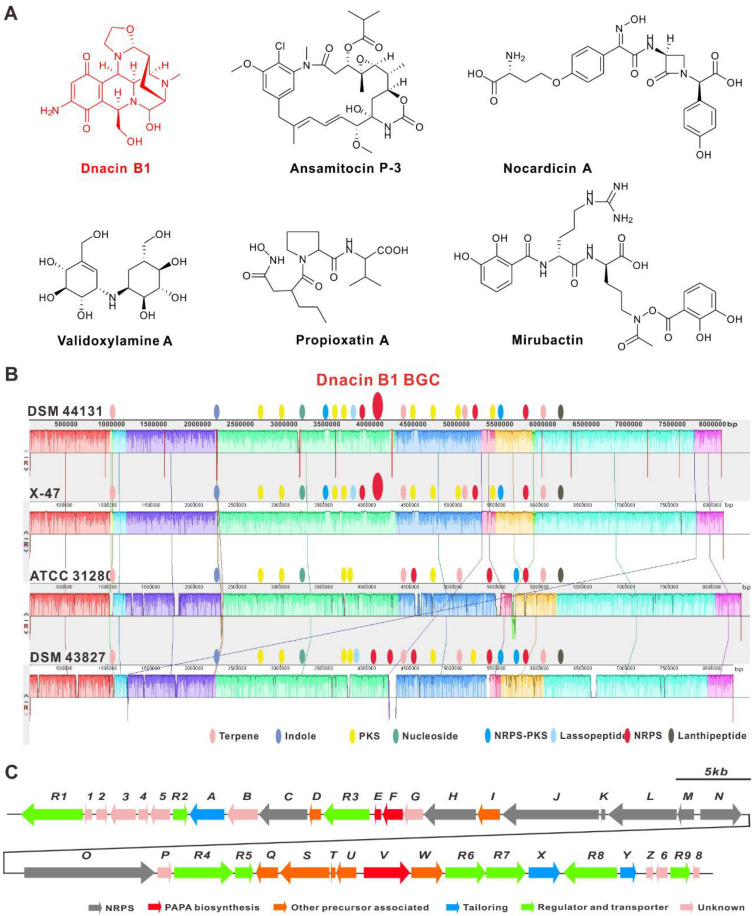
The diversified biosynthetic gene clusters located in the chromosome of *Actinosynnema* strains (DSM 44131^T^, X47, ATCC 31280 and DSM 43827^T^). (**A**) The biological natural products discovered from the genus *Actinosynnema*. (**B**) Comparative genome sequence alignment among the *Actinosynnema* strains performed by MAUVE, indicating the variations in secondary metabolite-encoded biosynthetic gene clusters (BGCs) and discovering the BGC of dnacin B1 in the genome of strain DSM 44131^T^ and strain X47. (**C**) Genetic organization of dnacin B1 BGC.

**Figure 2 molecules-25-04186-f002:**
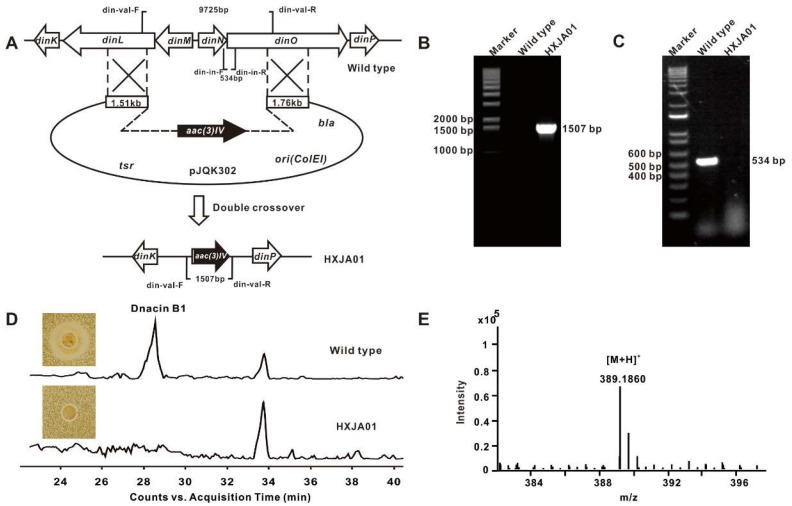
Inactivation of dnacin B1 BGC. (**A**) Construction of the dnacin B1 BGC inactivation mutant by homologous recombination. (**B**) PCR validation of wild type and mutant HXJA01. When using din-val-F/R as primers, the product of mutant HXJA01 was 1507 bp, whereas there was no product in wild type. (**C**) PCR analysis of wild type and mutant HXJA01. When using din-in-F/R as primers, the product of wild type was 534 bp, whereas there was no product in mutant HXJA01. (**D**) HPLC-MS and bioassay analysis of wild type and mutant HXJA01. (**E**) HR-ESI-MS analysis result of the dnacin B1.

**Figure 3 molecules-25-04186-f003:**
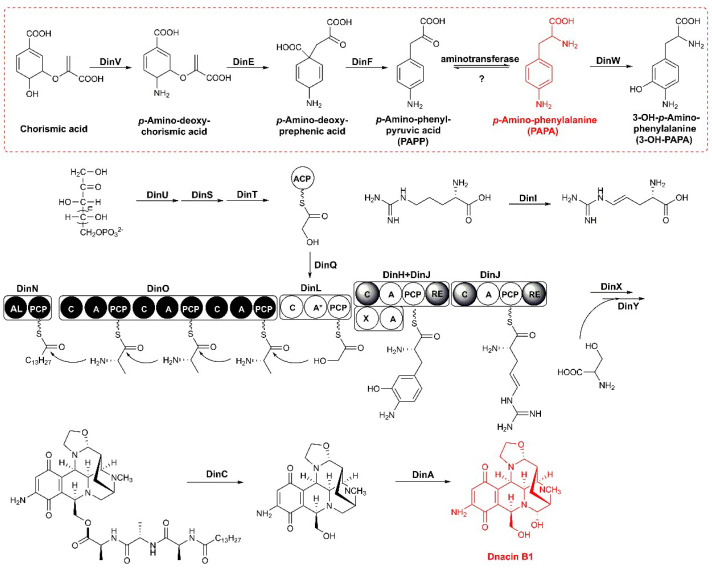
Proposed biosynthetic pathway of dnacin B1. The biosynthetic pathway of assumed precursor 3-OH-PAPA was presented in red dotted box.

**Figure 4 molecules-25-04186-f004:**
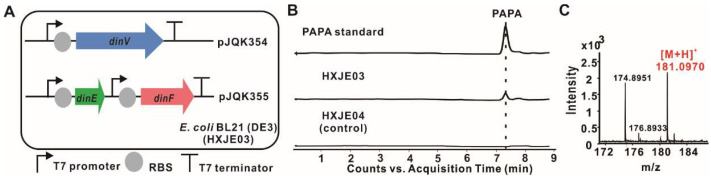
Heterologous expression of PAPA in *E. coli* BL21 (DE3). (**A**) Schematic representation of PAPA synthesis in engineered *E. coli* strain HXJE03. Genes *dinV*, *dinE* and *dinF* were all expressed under the control of T7 promoter. (**B**) HPLC-MS analysis of PAPA standard and the products of engineered *E. coli* strains (HXJE03 and HXJE04). (**C**) HR-ESI-MS analysis result of PAPA.

**Figure 5 molecules-25-04186-f005:**
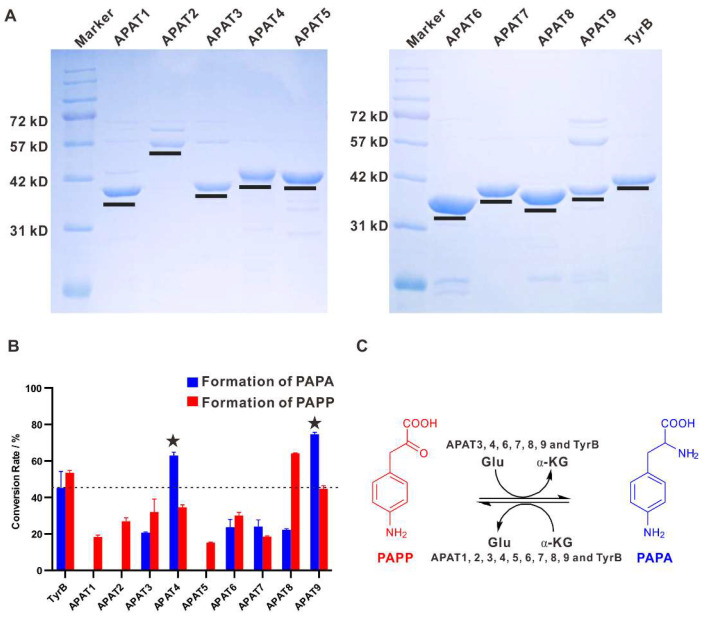
Expression and characterization of APATs and TyrB. (**A**) SDS-PAGE of nine purified APATs and TyrB expressed in *E. coli* BL21 (DE3). (**B**) Conversion rate analysis of recombinant nine APATs and TyrB in vitro. (**C**) The proposed reversible reactions of APATs and TyrB.

**Table 1 molecules-25-04186-t001:** Deduced functions of ORFs in the dnacin B1 biosynthetic gene cluster.

ORF	Number of Amino Acids	Proposed Function	Sequence Similarity (Protein, Origin)	% Identity/ Similarity	Accession No.
**DinR1**	951	ABC transporter	IQ63_41860, *Streptomyces acidiscabies*	64/75	KND24720.1
**ORF1**	117	unknown	SD37_10880, *Amycolatopsis orientalis*	55/75	ANN21703.1
**ORF2**	175	protease inhibitor	AC230_04780, *Streptomyces caatingaensis*	38/54	KNB54254.1
**ORF3**	395	histidine kinase	A9W97_15640, *Mycobacterium gordonae*	51/60	OBJ88517.1
**ORF4**	159	ATP-binding protein	RKT69099.1, *Saccharothrix variisporea*	45/51	DFJ66_2293
**ORF5**	306	epimerase	DI639_00155, *Leifsonia xyli*	74/83	PZO61411.1
**DinR2**	228	TetR family transcriptional regulator	STRAU_2618, *Streptomyces aurantiacus*	58/71	EPH44178.1
**DinA**	534	flavin adenine dinucleotide (FAD)-linked oxidase	NapU, *Streptomyces lusitanus*	64/75	AGD80628.1
**DinB**	471	gamma-aminobutyraldehyde dehydrogenase	MCBG_02966, *Micromonospora sp. M42*	53/65	EWM65833.1
**DinC**	741	peptidase	NapG, *Streptomyces lusitanus*	48/56	AGD80614.1
**DinD**	183	flavin mononucleotide (FMN) reductase	SsuE, *Streptomyces noursei*	58/67	WP_067345371.1
**DinR3**	707	regulatory protein	NapR3, *Streptomyces lusitanus*	55/68	AGD80615.1
**DinE**	103	4-amino-4-deoxychorismate mutase	PapB, *Streptomyces venezuelae*	34/53	BAD21142.1
**DinF**	307	4-amino-4-deoxyprephenate dehydrogenase	PapC, *Streptomyces venezuelae*	40/51	BAD21141.1
**DinG**	300	unknown	CLV43_102760, *Umezawaea tangerina*	47/57	PRY45195.1
**DinH**	796	adenylation domain	NapH, *Streptomyces lusitanus*	51/60	AGD80616.1
**DinI**	342	non-heme iron hydroxylase	NapI, *Streptomyces lusitanus*	63/75	AGD80617.1
**DinJ**	1482	NRPS	NapJ, *Streptomyces lusitanus*	68/79	AGD80618.1
**Module 5**		C-A-PCP-RE			
**DinK**	66	MbtH-like protein	NapK, *Streptomyces lusitanus*	73/84	AGD80619.1
**DinL**	1125	NRPS	NapL, *Streptomyces lusitanus*	54/64	AGD80620.1
**Module 4**		C-A-PCP			
**DinM**	252	thioesterase II	NapM, *Streptomyces lusitanus*	51/64	AGD80621.1
**DinN**	632	AMP-dependent synthetase and ligase	NapN, *Streptomyces lusitanus*	64/74	AGD80622.1
**Loading**		AL-PCP			
**DinO**	3049	NRPS	NapO, *Streptomyces lusitanus*	52/63	AGD80623.1
**Module 1**		C-A-PCP			
**Module 2**		C-A-PCP			
**Module 3**		C-A-PCP			
**DinP**	223	hypothetical protein	NapP, *Streptomyces lusitanus*	45/59	AGD80633.1
**DinR4**	909	ABC transporter	B0I31_10111, *Saccharothrix carnea*	56/67	PSL57800.1
**DinR5**	280	ABC transporter	CLV69_102810, *Yuhushiella deserti*	53/69	TDX97704.1
**DinQ**	336	3-oxoacyl-synthase III (KS)	NapE, *Streptomyces lusitanus*	72/80	AGD80612.1
**DinS**	754	Transketolase	NapD, *Streptomyces lusitanus*	59/68	AGD80611.1
**DinT**	76	acyl carrier protein (ACP)	NapC, *Streptomyces lusitanus*	63/75	AGD80633.1
**DinU**	306	transketolase	NapB, *Streptomyces lusitanus*	68/77	AGD80609.1
**DinV**	706	4-amino-4-deoxychorismate synthase	PapA, *Streptomyces venezuelae*	48/58	BAD21140.1
**DinW**	486	4-hydroxyphenylacetate 3-monooxygenase	EWI31_24745, *Streptomyces tsukubensis*	56/69	TAI42134.1
**DinR6**	613	ABC transporter	CLV43_102777, *Umezawaea tangerina*	70/78	PRY45212.1
**DinR7**	601	ABC transporter	DIU55_13605, *Firmicutes bacterium*	51/66	PZN68908.1
**DinX**	477	monooxygenase	NapA, *Streptomyces lusitanus*	69/76	AGD80608.1
**DinR8**	818	UV-repair protein	NapR1, *Streptomyces lusitanus*	75/84	AGD80606.1
**DinY**	244	methyltransferase	QncJ, *Streptomyces melanovinaceus*	47/65	AGD95052.1
**DinZ**	118	long-chain-fatty-acid--CoA ligase	NCTC13184_06982, *Nocardia africana*	60/81	SUA48430.1
**ORF6**	182	sugar O-acetyltransferase	E1091_12580, *Micromonospora fluostatini*	64/72	TDB92781.1
**DinR9**	304	LysR family transcriptional regulator	AFR_16085, *Actinoplanes friuliensis*	57/69	AGZ41499.1
**ORF7**	104	transposase	CLV43_10550, *Umezawaea tangerina*	56/61	PRY41292.1

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
