# Peer review of "p-Aminophenylalanine Involved in the Biosynthesis of Antitumor Dnacin B1 for Quinone Moiety Formation"

_molecules, 2020, doi:10.3390/molecules25184186_

Round 1

Author Response

Point 1: L138: NPRS → NRPS, L144: NPRS → NRPS

 Response 1: We are sorry for our carelessness, and have corrected the abbreviation.

Point 2: In L210-L212, it is described that pJQK354 was constructed from pETDuet and pJQK355 was constructed from pCDFDuet. However, the description is reversed for L404-407 (pJQK354 was constructed from pCDFDuet, pJQK355 was constructed from pETDuet). In Table S1, pJQK354 is pCDFDuet with dinV gene and pJQK355 is pETDuet with dinE and dinF gene. These are inconsistent.

Response 2: Thanks for the careful reading of the reviewer. We have checked it already in the manuscript.

Point 3: In Figure 5C, PAPA and α-KG are used as substrates in the reaction in which PAPP is produced, not PAPA and Glu. The description of L228-L229 and Figure 5C are inconsistent.

Response 3: We have carefully checked Figure 5C and revised the error accordingly.

Point 4: L316: “The purification of PAPA and PAPA were carried……”Is the description "PAPA and PAPA" "PAPP and PAPA" or "PAPA and PAPP"?  Please confirm.

Response 4: Actually “and PAPA” is redundant, and we have deleted this expression in the manuscript.

Reviewer 2 Report

In this paper, authors identified dancin B1 biosynthetic gene cluster from Actinosynnema pretiosum subsp. Auranticum strain DSM 44131. The authors analyzed whole genome of the DSM 11431 strain and analyzed the genome by comparing it with related strains (Actinosynnema) and analyzed the genome with antiSMASH. Taken together with culture condition optimization, transcriptional analysis and bioinformatic analysis, the authors discovered a candidate for the dancing B1 biosynthetic gene cluster. This cluster was homologous to naphthyridinomycin biosynthetic genes and the biosynthetic pathway of dancing B1 was proposed according to their homology. The gene inactivation experiment confirmed that the discovered cluster is actually responsible for biosynthesis of dnacin B1. Because the characteristic feature of dancin B1 is the usage of p-aminophenylalanine in its biosynthesis, the authors reconstruct biosynthesis of p-aminophenylalanine by the heterologous expression of dinV, dinE and dinF gene. The final step of p-aminophenylalanine biosynthesis (reductive transamination) seemed to be catalyzed by endogenous TyrB in E. coli. This was confirmed by in vitro analysis. Next to identified an enzyme catalyzing reductive transamination reaction in Actinosynnema, the authors produced recombinant putative aminotransferases in the genome of the DSM 11431 strain because the biosynthetic gene cluster do not have a gene encoding such an enzyme. As a result, two enzymes were indicated to be responsible for the reaction. Because the biosynthetic genes of dnacin have not been reported this paper should interest researchers in the biosynthetic field. 

Can authors predict the substrate specificities of A domains?
Or can mention how antiSMASH predict the substrate specificity?

The authors analyzed 9 APAT genes in the genome. Are these conserved in four Actinosynnema strains in Figure 1 or not? Is there any gene which only exist in DSM 44131 and X-47 strains?

"p-amino phenylalaine" in the title is better to be "p-amino-phenylalanine" or "p-aminophenylalanine"

Figure 3, please describe C and RE domains of DinJ in the figure.

Line 300. MutiGeneBlast should be MultiGeneBlast.

Author Response

Point 1: Can authors predict the substrate specificities of A domains? Or can mention how antiSMASH predict the substrate specificity?

 Response 1: The substrate specificities of A domains were predicted with online analysis website as mentioned in Materials and methods. Certainly, antiSMASH is also a powerful tool, and can precisely predict the substrate specificity of A domain in DinO. Considering the unique structure of dnacin B1, there are some untypical NRPS proteins like DinL and DinJ, of which the substrate specificities are unpredictable within general database. So this part of work is carried on by comparing with high homologous proteins in other gene clusters, e.g., NDM and QNC BGC.

Point 2: The authors analyzed 9 APAT genes in the genome. Are these conserved in four Actinosynnema strains in Figure 1 or not? Is there any gene which only exist in DSM 44131 and X-47 strains?

Response 2: All the nine APAT genes are found in the other three Actinosynnema strains in Figure 1, indicating that the aminotransferases in basic metabolism are also in charge of the PAPA biosynthesis. And we have supplemented this information in Discussion and Table S6.

Point 3:p-amino phenylalanine” in the title is better to be “p-amino-phenylalanine” or “p-aminophenylalanine”.

Response 3: Thanks for the correction. We have changed “p-amino phenylalanine” into “p-aminophenylalanine” in the title and keywords.

Point 4: Figure 3, please describe C and RE domains of DinJ in the figure.

Response 4: As recommended, we have described C and RE domains of DinJ in Figure 3.

Point 5: Line 300, MutiGeneBlast should be MultiGeneBlast.

Response 5: The mistake has been revised.

Reviewer 3 Report

The Author should consider also the following articles, recently published in Natural Product Research.

  1. Two novel ansamitocin analogs from Actinosynnema pretiosum. Siyu-Mao , Hong-Chen , Li-Chen , Chuanxi-Wang , Wei-Jia , Xiaoming-Chen , Huangjian-Yang , Wei-Huang & Wei-Zheng. Natural Product Research, Volume 27, 2013 - Issue 17.
  2. A new antitumour ansamitocin from Actinosynnema pretiosum. Guo-Zhu Wei , Lin-Quan Bai , Tao Yang , Juan Ma , Ying Zeng , Yue-Mao Shen & Pei-Ji Zhao. Natural Product Research, Volume 24, 2010 - Issue 12.

Author Response

Point 1: The author should consider also the following articles, recently published in Natural Product Research.

 Response 1: We sincerely appreciate the valuable comments. We have checked the two suggested articles carefully and added more references on L40 into the Introduction part in the manuscript.

  1. Wei, G. Z.; Bai, L. Q.; Yang, T.; Ma, J.; Zeng, Y.; Shen, Y. M.; Zhao, P. J. A new antitumour ansamitocin from Actinosynnema pretiosum. Nat. Prod. Res. 2010, 24, 1146-50.
  2. Siyu, M.; Hong, C.; Li, C.; Chuanxi, W.; Wei, J.; Xiaoming, C.; Huangjian, Y.; Wei, H.; Wei, Z. Two novel ansamitocin analogs from Actinosynnema pretiosum. Nat. Prod. Res. 2013, 27, 1532-6.